# Machine learning mathematical models for incidence estimation during pandemics

Oscar Fajardo-Fontiveros[1], Mattia Mattei[2], Giulio Burgio[2], Clara Granell[2], Sergio Gómez[2], Alex Arenas[2,3]*, Marta Sales-Pardo[1]*, Roger Guimerà[1,4]*

1 Department of Chemical Engineering, Universitat Rovira i Virgili, Tarragona, Catalonia, 2 Department of Computer Science and Mathematics, Universitat Rovira i Virgili, Tarragona, Catalonia, 3 Pacific Northwest National Laboratory, 902 Battelle Blvd, Richland, Washington, United States of America, 4 ICREA, Barcelona, Catalonia

* alexandre.arenas@urv.cat (AA); marta.sales@urv.cat (MS-P); roger.guimera@urv.cat (RG)

## Abstract

Accurate estimates of the incidence of infectious diseases are key for the control of epidemics. However, healthcare systems are often unable to test the population exhaustively, especially when asymptomatic and paucisymptomatic cases are widespread; this leads to significant and systematic under-reporting of the real incidence. Here, we propose a machine learning approach to estimate the incidence of a pandemic in real-time, using reported cases and the overall test rate. In particular, we use Bayesian symbolic regression to automatically learn the closed-form mathematical models that most parsimoniously describe incidence. We develop and validate our models using COVID-19 incidence values for nine different countries, confirming their ability to accurately predict daily incidence. Remarkably, despite the differences in epidemic trajectories and dynamics across countries, we find that a single model for all countries offers a more parsimonious description and is more predictive of actual incidence compared to separate models for each country. Our results show the potential to accurately model incidence in real-time using closed-form mathematical models, providing a valuable tool for public health decision-makers.

**Data Availability Statement:** The data used in this study are available from https://ourworldindata.org/ (Belgium, Canada, Croatia, France, Hungary, India, Italy, and United Kingdom), and https://dadescovid.

## Author summary

In pandemic situations, adopting timely, effective preventive measures requires accurate and real-time estimates of the incidence of the disease. However, estimates of incidence are typically biased and incomplete. Here, we propose a machine learning method that allows us to discover mathematical models that accurately estimate real incidence from readily available measures, such as the number of cases detected and the number of tests administered on a given day. Contrary to heuristic approaches, which are forced to make bold assumptions about the models, our approach automatically selects the most parsimonious models from the data, with very mild assumptions about the structure of the biases of incidence estimates based on incomplete testing. Our models outperform others at predicting real incidence. Additionally, we find that the same models can be applied to

cat/descarregues (Catalonia). Code for the BMS is open and freely available from https://bitbucket.org/rguimera/machine-scientist/.

**Funding:** This research was supported by projects PID2022-142600NB-I00 (M.SP. and R.G.) and PID2021-128005NB-C21 (A.A., C.G. and S.G) from MCIN/AEI/10.13039/501100011033; by project 2021SGR-633 from the Government of Catalonia (all authors); and by project 2023PFR-URV-00633 from Universitat Rovira i Virgili (all authors). A.A. GB, SG and CG acknowledge support from project CREXDATA no. 101092749 from the European Union's Horizon Europe Programme, and from project no.\ 220020325 from the James S. McDonnell Foundation. AA also acknowledges the Joint Appointment Program at Pacific Northwest National Laboratory (PNNL). PNNL is a multi-program national laboratory operated for the U.S. Department of Energy (DOE) by Battelle Memorial Institute under Contract No. DE-AC05-76RL01830. This project has also received funding from the European Union's Horizon 2020 research and innovation programme under the Marie Skłodowska-Curie grant agreement no. 945413 (M. M.). The funders had no role in study design, data collection and analysis, decision to publish, or preparation of the manuscript.

**Competing interests:** The authors have declared that no competing interests exist.

describe nine different countries with very different socioeconomic characteristics and epidemic dynamics.

## Introduction

The control of an epidemic hinges on having accurate estimates of the incidence of the infection, that is, the number of newly infected individuals in the population. A reliable method for obtaining accurate and consistent estimates of incidence involves testing large, random samples of the population regularly [1]. However, healthcare systems often lack the capacity to perform such large-scale testing campaigns. This limitation was evident during the COVID-19 pandemic, where testing samples were typically small and predominantly focused on symptomatic individuals [2]. Given the high number of asymptomatic and paucisymptomatic cases, this approach results in a considerable underestimation of the actual incidence of infection.

Among other major effects on epidemic control, under-detection of cases leads to silent spreading, and makes it difficult to estimate key epidemiological parameters such as the basic reproduction number $R_0$ and the time-varying reproduction number $R_t$ [3–5]. As a consequence of the latter, under-reporting compromises attempts to forecast incidence and to evaluate the effect of past and future control measures, which is further aggravated by the fact that under-reporting is not constant, but changes with time and geography, because of changes in epidemic response and testing practices [5–8].

A potential strategy to achieve more accurate estimates of incidence that correct for under-reporting is to use compartmental models calibrated against time series of reported cases, hospitalizations, and/or fatalities [8–15]. Yet, while these models can describe the entire temporal evolution of an epidemic and the impacts of various interventions *ex post*, they are unable to provide real-time estimates of incidence. Similarly, reconstruction of real incidence from some form of deconvolution of the time series of fatalities [6, 16] is only possible retrospectively.

A more effective strategy for real-time estimation involves the utilization of semi-empirical and data-driven models. These models establish functional correlations between the incidence of a disease and daily accessible metrics, such as reported cases and the number of tests conducted [17–19]. Along these lines, here we introduce a data-driven machine learning approach to estimate the daily incidence of a pandemic, using only reported cases and the test rate across the population. Our method differs from previous models that assumed and imposed specific functional relationships between the actual incidence, detected cases, and test rate. Instead, we employ automated equation discovery [20, 21] and, in particular, Bayesian symbolic regression [22–24], to automatically derive parsimonious closed-form mathematical models capturing this relationship.

We train and validate our models using reconstructed COVID-19 incidence data from nine different countries. In particular, we train the Bayesian symbolic regression algorithm using these COVID-19 incidence time series from each of the nine countries over a four-month period. We then validate our models in terms of their ability to precisely estimate unobserved daily incidences beyond the training data, in conditions that emulate real-time prediction of incidence during the following two months. Moreover, and in contrast with standard machine learning approaches, ours allow us to train and compare: (i) specific models for each country; *versus* (ii) a single, unified mathematical model that represents daily incidences across all countries. Contrary to what one may expect from the heterogeneity of testing practices and from the variability in detection rates, even within single countries [6–8], we conclude that a unified model, tailored with country-specific parameters, offers a more parsimonious and

predictive representation of data compared to using distinct models for each country. From a practical perspective, our findings demonstrate that our models are not only precise in estimating real-time incidence rates but also readily applicable tools for public health policymakers.

## Results

### Machine learning models to estimate daily unreported cases

We aim to provide a (nearly) real-time estimation of the incidence $I(t)$, defined as the number of newly infected individuals in a population on day $t$. We write $I(t)$ in terms of the number $C(t)$ of reported cases on that day, which, assuming that individuals are not tested more than once per day, coincides with the number of positive tests on that day. Therefore, we have

$$I(t) = \frac{1}{\rho} \times C(t),\tag{1}$$

where $\rho$ is the reporting (or detection) rate, and the factor $1/\rho$ represents how many real new cases there are on day $t$ for each reported case on the same day. Given that $C$ is a known quantity, the challenge is to obtain an expression for $\rho$ as a function of known daily variables. In particular, we expect $\rho$ to have a non-trivial dependence on the fraction $T$ of the population tested. Indeed, considering for a moment a uniform, constant infection period, in the absence of testing bias towards infected individuals $\rho$ would be proportional to $T$ (the constant of proportionality being the inverse of the infection period). For the same $T$ and $I$, if testing is instead biased, then $C$ is higher and must be compensated by a larger $\rho$. The bias is expected to be especially strong for small test samples, mostly consisting of infected individuals, while gradually disappearing for increasingly larger samples.

These qualitative behaviors can be accounted for by a simple power-law $1/\rho \propto T^n$, with $n \leq 0$, as proposed by Chiu *et al.* [18] The closer $n$ is to zero, the larger the bias. This model has the desirable limiting behaviors: (i) $\rho(T = 0) = 0$, so that $C = 0$ when there is no testing; (ii) $\rho(T = 1) = 1$, so that $C = I$ when the whole population is tested and all cases are reported. (Strictly speaking, in the limit $T = 1$ one would need to exclude from testing individuals that were already reported positive on previous days; otherwise, $C$ would coincide with the prevalence, rather than the incidence. Also strictly speaking, reporting is delayed with respect to infection. If the times between infection and detection are narrowly distributed, the infection and detection curves are just shifted in time; otherwise, there is a smoothing of the detection curve compared to the infection curve, in addition to the shift.) Following this reasoning, Chiu *et al.* [18] estimated the fraction of non-detected infections using constant, country-dependent exponents $n$. Their estimates were more accurate than those obtained from data-driven epidemiological models.

Here, we argue that bias, and hence under-reporting, cannot be entirely accounted for by test coverage. In fact, even for the same number of administered tests, the bias may change in response to the current state of the epidemic, making it plausible that the factor $1/\rho$ depends also on the reported cases $C(t)$. Building on these observations, we propose a more general model $1/\rho = T^{n(T,C)}$, where the exponent $n(T, C)$ is allowed to be a function of $T$ and $C$. (Note that the reporting exponent $n$ is not a direct measure of the bias when, as it happens in reality, the infection period is distributed. Indeed, in that case we no longer have a reference value for $n$ indicating unbiased sampling. Nonetheless, for a fixed test coverage, it is still true that the smaller is $n$, the lower is the bias.) Rather than postulating a certain functional dependency of the reporting exponent $n(T, C)$, we propose an automated method to learn this dependency from data.

Indeed, we assume that, for a limited training period, we have time series $T(t)$ and $C(t)$, and the ground truth incidence $I(t)$. The daily value of the reporting exponent $n$ for such dataset can be estimated using Eq (1) as

$$n(t) = \frac{\log I(t) - \log C(t)}{\log T(t)} \; .\tag{2}$$

With $n(t)$ calculated in this way, we aim to find a closed-form mathematical model $n = n(T, C)$ that describes the dependency of $n$ on $T$ and $C$ (and, thus, that of the reporting rate $\rho$). (By closed-form we mean a model that can be expressed in terms of standard operations; for example, $n = (C/T)^2$ or $n = \exp(T + C)/T$; see Methods.) To do this, we use a Bayesian symbolic regression approach known as the Bayesian machine scientist (BMS; Methods) [22]. Given a dataset $D$, the BMS samples models $n(T, C)$ from the posterior distribution over models

$$P(n(T,C)|D) = \frac{\exp\left[-\mathcal{L}(n(T,C))\right]}{Z(D)},\tag{3}$$

where $\mathcal{L}(n(T,C))$ is the description length [25] of the model and the dataset [22]. The description length measures the number of nats needed to encode both the dataset $D$ and the model $n(T, C)$. According to Eq (3), the most plausible model is the one with minimum description length, that is, the one that provides the most compressed description of the data. This method is consistent (asymptotically, it selects the true generating model with probability approaching one) and it outperforms other symbolic regression approaches [22, 24].

## The BMS learns the correct reporting rate from synthetic data

We start by validating our approach on synthetic data. In particular, we use a compartmental epidemic model to simulate four different spreading scenarios (see Methods for details and exact parametrizations): (i) a scenario of decreasing incidence after a peak; (ii) a scenario of increasing incidence leading to a peak; (iii) a scenario with a complete wave; and (iv) a scenario with two consecutive waves. For each one of these scenarios, we simulate four testing and detection models (Methods). In the first three models, testing grows indefinitely with incidence; in the fourth model, testing initially grows with incidence but it eventually saturates, thus simulating a more realistic setting with limited testing capabilities. With regards to reporting, in the first two models we assume constant reporting exponents ($n = -1$ and $n = -2$, respectively) as in Ref. [18], whereas in models three and four we assume that the reporting exponent explicitly depends on $C$. Our experimental design thus results in 16 different validation experiments (Fig 1).

In all these experiments, we simulate 200 days; we use the first 40 of them for training and the remaining 160 for testing. In all cases, the BMS is able to identify the correct closed-form model for the reporting experiment $n(C, T)$ as the most plausible model, and thus provides optimal estimates of real incidence (Fig 1).

## The reporting exponent is not constant in empirical data

Having validated our approach on synthetic data, we move on to real data on the daily number of positives $C(t)$ and tests $T(t)$ during the second wave of the COVID-19 epidemic (August 2020 to January 2021, before the first massive vaccination campaigns), and from an estimate of the real incidence $I(t)$ during the same period. We obtain equivalent data for nine different countries: Belgium, Canada, Catalonia, Croatia, France, Hungary, India, Italy, and the United Kingdom (UK). We estimate the incidence $I(t)$ by deconvolving the daily number of fatalities (Methods) [16], which implicitly assumes that the infection fatality rate did not change

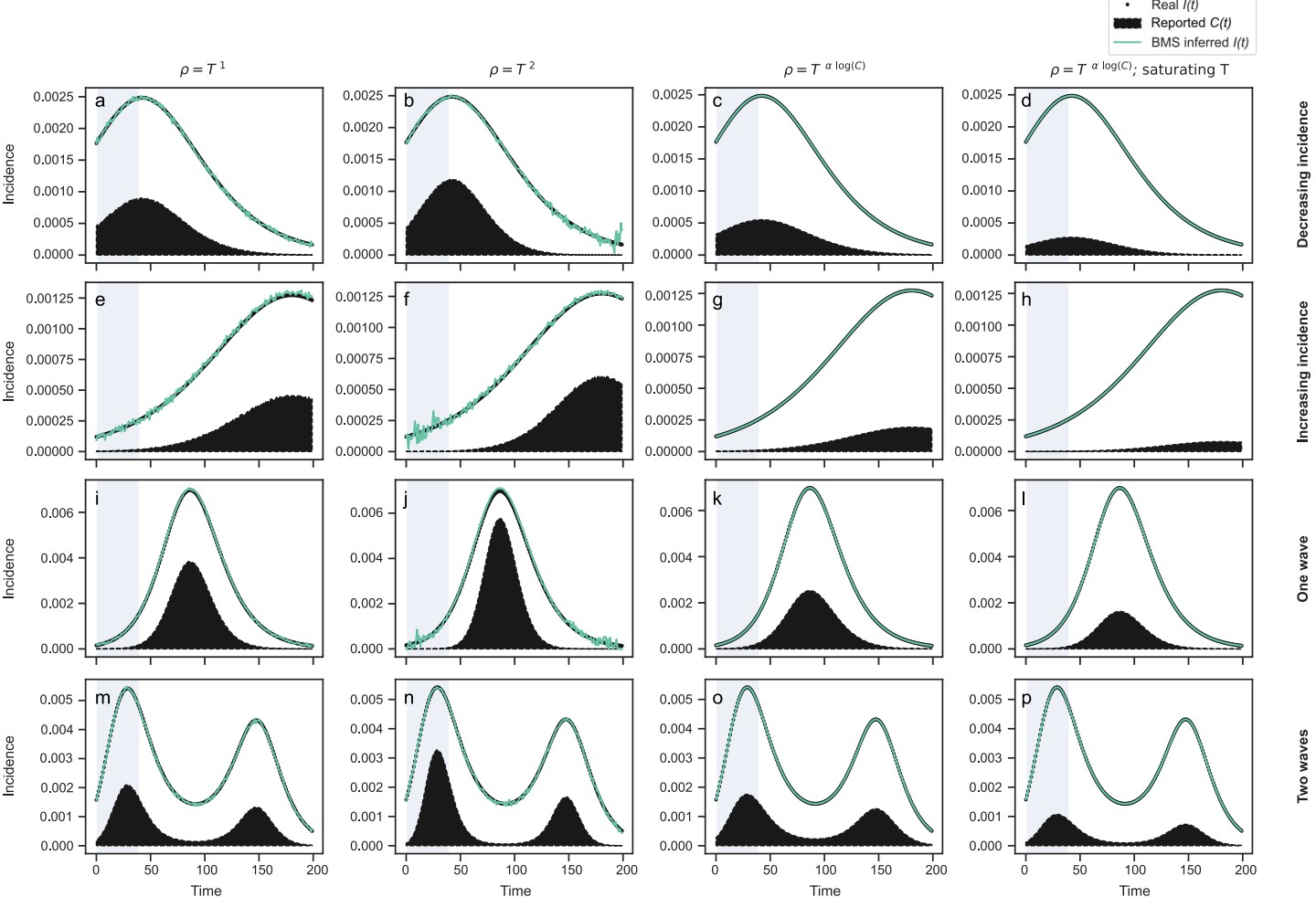

**Fig 1. Estimation and prediction of the incidence $I(t)$ in synthetic data.** We consider four simulated epidemic scenarios: (a-d) decreasing incidence after a peak; (e-h) increasing incidence leading to a peak; (i-l) a complete wave; and (m-p) two consecutive waves. For each of these scenarios, we consider four testing and reporting models (see Methods for details): constant reporting exponent $n = -1$ and testing that grows indefinitely with incidence (left column); constant reporting exponent $n = -2$ and testing that grows indefinitely with incidence (center-left column); variable reporting exponent $n = -\alpha \log C$ and testing that grows indefinitely with incidence (center-right column); variable reporting exponent $n = -\alpha \log C$ and testing that saturates at high incidence values (right column). Black dots represent the ground truth incidence $I(t)$; the black area represents the reported cases $C(t)$; and the green line represents the incidence estimated by the BMS after obtaining a closed-form expression for the reporting exponent $n(C, T)$. We simulate 200 days; we use the first 40 day for training (shaded area) and the last 160 for testing (white area). In all cases, the BMS is able to identify the correct closed-form model for the reporting experiment $n(C, T)$, and thus provides optimal estimates of real incidence.

significantly during the period considered (as opposed to what happened after vaccination started). However, we note that, in practice, one may obtain estimates of $I(t)$ by other methods such as sentinel testing, serological studies, syndromic surveillance or any other; and that our methodology does not hinge on the particular method used to estimate incidence for training.

We assume that each country $i$ has its own exponent, but train the BMS using two different hypotheses in this regard. First, we assume that each country $i$ is described by an entirely different functional form $n_i(T, C)$. To do that, we sample models from the posterior distribution $p(n_i|D_i)$ for each country $i$. Second, we assume that $n_i(T, C)$ has the same functional dependency $n(T, C)$ for all countries, but allow for model parameters $\theta$ to be country specific so that $n_i(T, C) = n(T, C; \theta_i)$. In this case, we sample models from the posterior distribution $p(n|D)$, where $D = \{D_i\}$ includes all country-specific datasets (Methods) [23].

In both cases, we obtain models with the BMS using the first four months of data (August 2020 to November 2020). We report results for: (i) the median of the whole ensemble of models sampled by the BMS (ensemble model); (ii) the single closed-form model that most resembles the median ensemble prediction (median predictive model). To evaluate the predictive power of the proposed models, we then use those models to make predictions during the last two months of the data (December 2020 and January 2021, which were not used for training), and measure the mean absolute error (MAE) of the predictions. All reported values for the MAE are on these test data only. Note that, after training, the reporting exponent $n(t)$, the reporting rate $\rho(t)$, and real incidence $I(t)$ are all estimated from $T(t)$ and $C(t)$ alone, which were typically available daily during the COVID-19 pandemic, with little delay or error in the countries considered. Therefore, our empirical validation closely emulates predictions made in real time during the two test months following training. In real settings, delays in reporting and "occurred but not reported events" could be corrected for, if necessary, using well-established methods [26]. Note also that $C(t)$, the empirical $I(t)$ and the different model estimates for $I(t)$ are not consistently and significantly shifted with respect to each other; therefore, we do not need any additional correction for time delays. (In some situations in which delays are larger, it may be necessary to shift $I(t)$ or $C(t)$ prior to training, and take this shift into account when making predictions.).

We start by studying whether the reporting exponent $n$ is well approximated by a constant value, as previously assumed by Chiu *et al.* [18]. As we show in Fig 2, we find that $n$ fluctuates considerably in time in the period considered, especially in the early months. Except for India, the overall tendency of the exponent is to become more negative with time. Besides the fact that it is visually apparent that the reporting exponent changes in time, we provide two additional quantitative arguments that confirm this observation. First, for the training period August through November, 2020, the BMS finds models that have shorter description length than the $n$ = const. model. These models are more plausible and more compressive of the data, and therefore are objectively better models for the training period. Second, the models proposed by the BMS are more predictive on the test data, that is, for the extrapolation to months December 2020 through January 2021 (Fig 2). Indeed, for seven of the nine countries considered, all BMS models are better than the constant model. For another country, Hungary, the best model is still a BMS model, but the constant model is better than some BMS models. Canada is the only country for which the BMS does not find a model for the exponent that is better than a constant. Note, however, that for Canada $n(t)$ happens to be quite constant in the test period, which explains why the constant model is the most compressive and also explains the fact that all models make excellent predictions with very small error. Had we taken a different test period (for example, August through September, 2020) the prediction error of the constant model would have been larger also for Canada.

## Variable reporting exponents lead to accurate predictions of reporting rate and incidence

The closed-form models $n(T, C)$ obtained by the BMS for the reporting exponent allow us to estimate the reporting rate $\rho(t)$ and, from Eq (1), the daily incidence $I(t)$. As in the previous section for the exponent $n$, we use the models obtained by the BMS to make predictions for the reporting rate (Fig 3) and the incidence in the test months (December 2020 through January 2021) (Fig 4). As before, we find that the models obtained with the BMS lead to more accurate predictions for the reporting rate and the incidence (as measured by the prediction MAE) for all countries except Canada, where the constant exponent performs slightly better despite

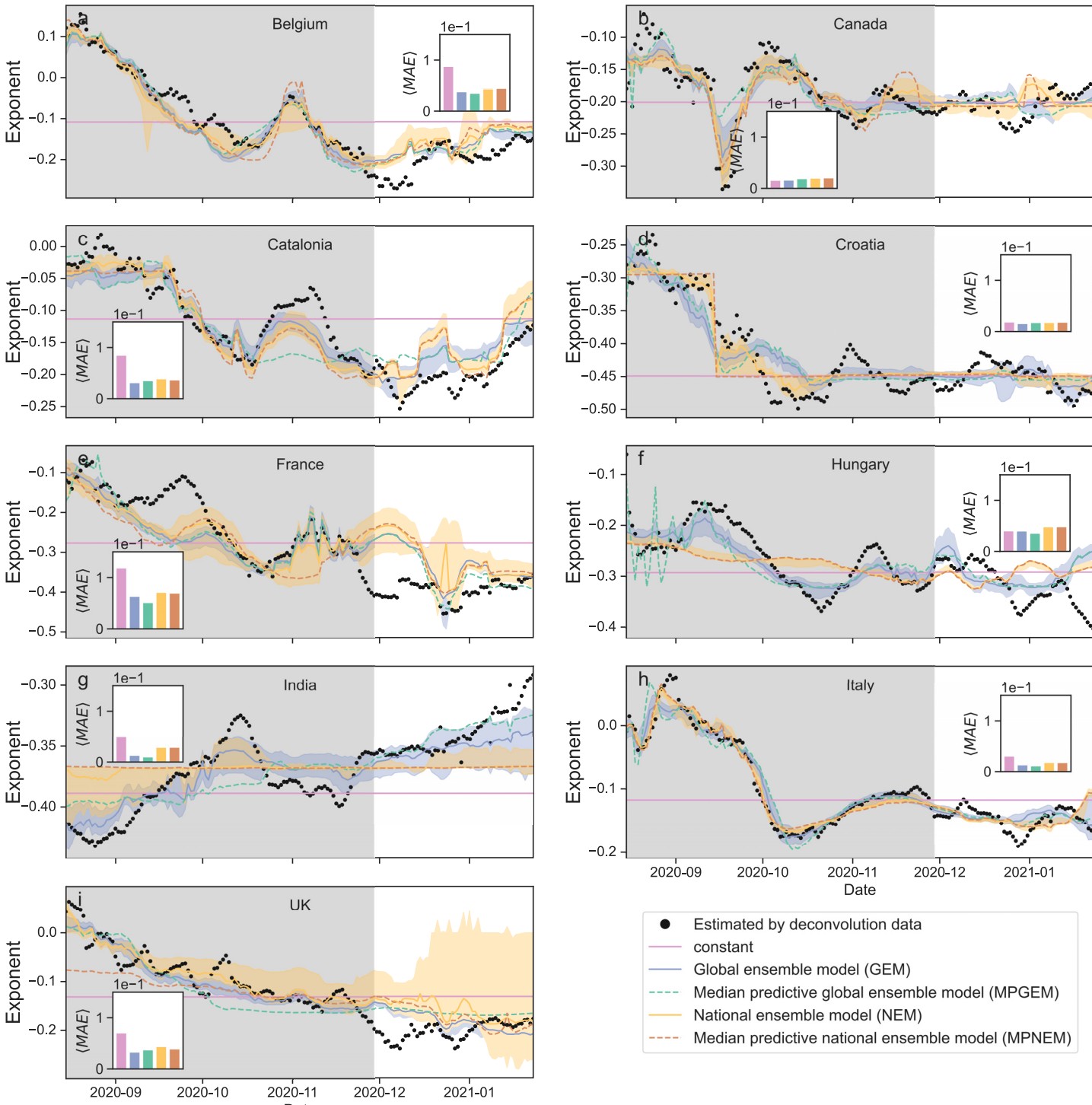

**Fig 2. Estimation and prediction of the reporting exponents $n(t)$ for different countries.** (a) Belgium; (b) Canada; (c) Catalonia; (d) Croatia; (e) France; (f) Hungary; (g) India; (h) Italy; (i) United Kingdom. The exponent determines the relationship among the real incidence $I$, the reported cases $C$ and the test rate $T$. For each country we show the estimated exponent, as well as the following models (see Methods): (i) the constant exponent model (solid pink line); (ii) the median of the ensemble of models obtained by the BMS (blue and orange solid lines), both when we model all countries together (blue: global ensemble model, GEM) and when we obtain individual models for each country (orange: national ensemble model, NEM); (iii) the median predictive model (dashed lines) obtained by the BMS, both when we model all countries together (green: median predictive global ensemble model, MPGEM) and when we obtain individual models for each country (brown: median predictive national ensemble model, MPNEM). Shaded regions indicate the interquartile range for model ensembles. We obtain the models using only data from August to the end of November, 2020 (shaded region). We then use those models to predict on the test data, December 2020 through January 2021. The insets show the mean absolute error

(MAE) of each model on the test data. All MAE are displayed using the same scale, so that one can see which countries are easier to predict and which are more difficult. For all countries except Canada, the most accurate model is one of the BMS models.

the fact that all models are very accurate. As mentioned earlier, this is due to the fact that, in Canada, $n$ is fairly constant for the duration of our test period.

## A single model for all countries is more parsimonious than country-specific models

Up to now, we have shown that mathematical models obtained with the BMS lead to accurate estimations and predictions of the reporting exponent, the reporting rate, and the incidence. In general, these predictions are more accurate than those provided by the model that assumes a constant exponent. However, it is also interesting to compare models obtained by the BMS to one another.

As noted earlier, we use the BMS to obtain two families of models. In the first family, a different set of models $n_i(T, C)$ is obtained for each country. This means that models of each country can have completely different mathematical forms and dependencies on $T$ and $C$. In the second family, the same set of models $n(T, C; \theta_i)$ is used for all of the countries, allowing only the parameters $\theta_i$ to change from country to country. If testing dynamics were radically different in each country, the family of country-specific models may be expected to be more predictive. Otherwise, a global model may be applicable to all countries. In this case, training concurrently with the data of all countries should lead to more accurate predictions, since information about each country is actually used by all other countries, thus resulting in more robust models.

Consistent with the latter scenario, the models obtained by considering all countries together yield more accurate predictions than country-specific models (see Figs 2, 3 and 4). In other words, a single model (that is, a single functional dependency on $T$ and $C$) for all countries is more parsimonious than one model for each country. This result is perhaps surprising, considering the important differences in reporting rate that have been found even within regions of the same country [6–8]. It suggests that sampling bias and under-reporting emerge from mechanisms that are common to all countries, raising questions about the nature of such mechanisms. We speculate that such mechanisms may be related to: (i) the availability (and limitations of) the testing infrastructure; (ii) the procedures used to select individuals for testing, which are typically biased towards those individuals that are more likely to be infected; and (iii) social and political pressures arising from pandemic states, and to collective public reactions to such states. From a more practical standpoint, the possibility to parametrize all countries within a single model is an important simplification. This can be a crucial advantage when it comes to rapid and widespread implementation, especially in situations where time is of the essence, such as during an outbreak.

## Discussion

In a post-COVID world, where the crisis has receded, it remains imperative to remember the lessons learned from it. Lack of reliable real-time incidence and prevalence estimators was a critical handicap during the COVID-19 pandemic. This deficiency compromised the timely implementation of interventions, often resulting in delayed actions, extended periods of social distancing, and other stringent measures, and diminishing their overall effectiveness. In this context, our research posits that employing machine learning, specifically Bayesian

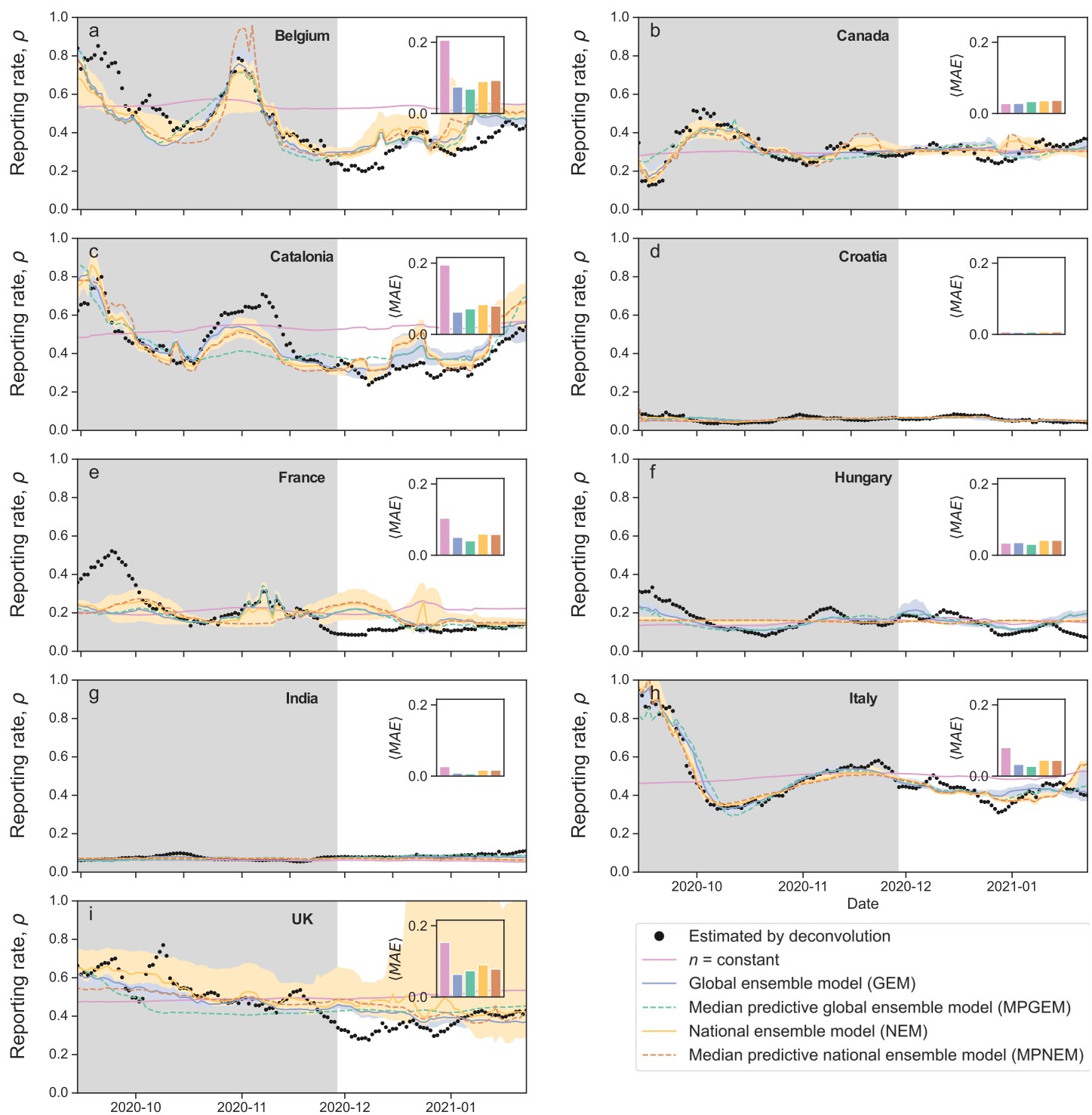

**Fig 3. Estimation and prediction of the reporting rate $\rho(t)$ for different countries.** (a) Belgium; (b) Canada; (c) Catalonia; (d) Croatia; (e) France; (f) Hungary; (g) India; (h) Italy; (i) United Kingdom. The reporting rate is modeled as $\rho = T^{-n}$ from the modeled reporting exponent $n(T, C)$ and the observed cases $C$ and testing rate $T$. For each country we show the estimated repoprting rate $\rho(t)$ (black dots), as well as the following models: (i) the constant exponent model (solid pink line); (ii) the median of the ensemble of models obtained by the BMS (blue and orange solid lines), both when we model all countries together (blue: global ensemble model, GEM) and when we obtain individual models for each country (orange: national ensemble model, NEM); (iii) the median predictive model (dashed lines) obtained by the BMS, both when we model all countries together (green: median predictive global ensemble model, MPGEM) and when we obtain individual models for each country (brown: median predictive national ensemble model, MPNEM). Shaded regions indicate the interquartile range for model ensembles. We use the same models as in Fig 2,

obtained using only data from August to the end of November, 2020 (shaded region). We then use those models to predict reporting rate on the test data, December 2020 through January 2021. The insets show the mean absolute error (MAE) of each model on the test data. All MAE are displayed using the same scale, so that one can see which countries are easier to predict and which are more difficult. For all countries except Canada, the most accurate model is one of the BMS models.

symbolic regression, to forecast the incidence of infectious diseases could be a transformative advancement.

Absent a comprehensive approach for real-time, large-scale population monitoring, we must rely on proxies to estimate incidence. Throughout the pandemic, the routine monitoring of reported cases in most countries proved inadequate for accurately gauging the true incidence of infection. This shortfall was due to logistic limitations, and the inability to detect asymptomatic individuals and those who avoided testing for various reasons. Consequently, the reported case figures necessitate correction to reflect the true incidence accurately. Similar shortcomings and their implications have been identified and discussed before in the context of COVID-19 and other diseases [6–8]. In our study, we have used Bayesian symbolic regression to develop mathematical models for this necessary correction. These models incorporate the number of tests conducted daily and the reported cases, enabling more precise estimations of the actual disease incidence beyond the training period. Although our analysis has focused on COVID-19, it is completely general and it could be used for any future epidemic, provided that incidence can somehow be estimated for a given training period, and testing and detection data are reported daily.

Our models are significantly more accurate than previous attempts to use simple heuristic functional forms. Additionally, our models are formulated as closed-form mathematical expressions, characterized by elementary functions and a limited number of parameters. This structure contrasts with the complex models typically derived from other machine learning methods, such as deep learning, and potentially facilitates ease of deployment and usability for public health experts. Along these lines, we discovered that a single model, adjustable for each country, is more parsimonious and yields more accurate incidence estimations than separate models for each country. This somewhat unexpected finding suggests a degree of uniformity in national responses to specific epidemiological situations, meriting further in-depth exploration. This insight, a direct result of symbolic regression's capacity to generate closed-form models, could not have been extracted from standard machine learning techniques.

However, we have not yet been able to use symbolic regression to its full potential. As we have discussed, the closed-form models we obtain are predictive, and they provide some degree of insight (the reporting exponent is not constant and depends on the state of the epidemics; one model for all countries is more parsimonious than a different model for each country); but they are not easily interpretable otherwise. Despite our efforts to interpret the resulting models, we have not been able to identify defining features or terms that could provide more precise insight about the mechanisms responsible for under-reporting. This is somehow expected, considering that they capture the behavior of countries as different as, for example, Canada and India; but we expect that future work will be able to propose simple ansatz for such models, and progress by restricting the search space of models, and/or by comparing simple plausible models to those identified by symbolic regression.

We also expect that further analyses, perhaps with new data or different time windows, may reveal additional insights. Indeed, here we have restricted our analysis to a single epidemic scenario (COVID-19 during the second half of 2020), and we cannot rule out the possibility that the identified models are somehow "overfit" to this scenario. Future work may try to apply the same approach to different scenarios simultaneously, and perhaps then simpler models, corresponding to more general mechanisms, may become apparent.

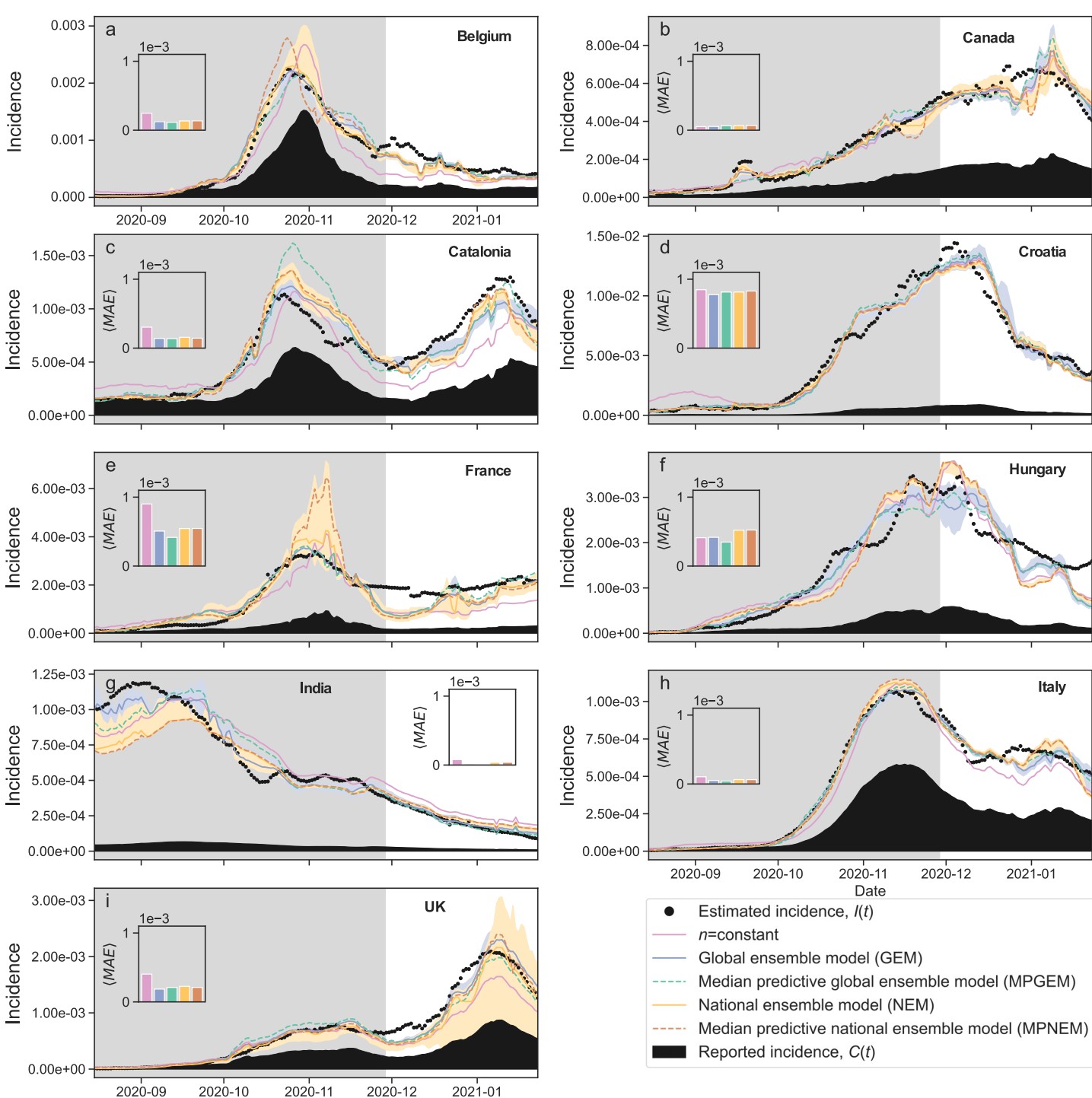

**Fig 4. Estimation and prediction of incidence $I(t)$ for different countries.** (a) Belgium; (b) Canada; (c) Catalonia; (d) Croatia; (e) France; (f) Hungary; (g) India; (h) Italy; (i) United Kingdom. The incidence is estimated as $I = T^n \times C$ from the modeled reporting exponent $n(T, C)$ and the observed cases $C$ and testing rate $T$. For each country we show the estimated incidence $I(t)$ (black dots) and the reported cases $C(t)$ (black area), as well as the following models: (i) the constant exponent model (solid pink line); (ii) the median of the ensemble of models obtained by the BMS (blue and orange solid lines), both when we model all countries together (blue: global ensemble model, GEM) and when we obtain individual models for each country (orange: national ensemble model, NEM); (iii) the median predictive model (dashed lines) obtained by the BMS, both when we model all countries together (green: median predictive global ensemble model, MPGEM) and when we obtain individual models for each country (brown: median predictive national ensemble model, MPNEM). Shaded regions indicate the interquartile range for model ensembles. We use the same models as in Fig 2, obtained using only data from August to the end of November, 2020 (shaded region). We then use those models to predict incidence on the test data, December 2020 through January 2021. The insets show the mean absolute error (MAE) of each model on the test data. All MAE are displayed using the same scale, so that one can see which countries are easier to predict and which are more difficult. For all countries except Canada, the most accurate model is one of the BMS models.

In any case, we think that our findings hold promise for future epidemic surveillance and management. They serve as a vital reminder that the insights gleaned from the COVID-19 pandemic must inform our approaches to future health crises.

## Materials and methods

### Data

The data consists in the daily number of positive tests, tests made, and reported deaths from 1 August 2020 to 31 January 2021 (i.e., during the second wave of the COVID-19 epidemic) for 9 different countries: Belgium, Canada, Catalonia, Croatia, France, Hungary, India, Italy, and United Kingdom. These countries were chosen according to the availability and the completeness of data in the period we study. These data are publicly available at https://ourworldindata.org/; the details and the methodology for the data collection are described in Hasell *et al.* [27]. For the same period, we also use the data for Catalonia (Spain) collected by the Generalitat de Catalunya and available at https://dadescovid.cat/descarregues. For each data stream (daily number of reported cases, tests and deaths for each country), we use a 7-day moving average, that is, the average computed from day $t - 6$ to day $t$.

To infer the number of daily infections from the reported deaths as described in the next section, we used the estimate of the infection fatality rate (IFR), that is the probability that an infected person dies, provided the World Health Organization (WHO) [28] and obtained by studies of the seroprevalence in the countries we consider (see Table 1).

### Estimating real incidence via deconvolution of fatalities

We used daily reported deaths to construct the *ground truth* against which to train and compare our data-driven models. Daily fatalities are, in general, informative about the infections occurred in previous days. However, the time elapsed between infection and death (time-to-death) depends on several factors and, as a result, can be highly variable among patients. According to Garcia *et al.* [29] and Linton *et al.* [30] a good choice for the time-to-death distribution is a lognormal distribution with a mean value of 20.1 days and a median equal to 18.8. These estimates were based on data about the first wave of the COVID-19 epidemic in early 2020. It is reasonable to believe that this amount of days may have slightly changed during the second wave in the late 2020. Faes *et al.* [31] stated that over the course of the first wave, the length of stay in hospital decreased two days on average from the onset of the first wave to the later stages of the first wave. Therefore, we choose to adopt a lognormal distribution with mean and median equal to 18.1 and 16.8 days, respectively. Nonetheless, we performed a

**Table 1. Population and infection fatality rate (IFR) [28] for the considered countries.**

| Country | Population | IFR |
|---|---|---|
| Belgium | 11,660,799 | 0.0087 |
| Canada | 38,213,305 | 0.0059 |
| Catalonia | 7,697,069 | 0.0092 |
| Croatia | 4,070,275 | 0.0014 |
| France | 65,476,462 | 0.0030 |
| Hungary | 9,626,188 | 0.0054 |
| India | 1,399,362,910 | 0.0007 |
| Italy | 60,337,813 | 0.0120 |
| UK | 68,390,408 | 0.0093 |

sensitivity analysis considering also 20.1 and 16.1 days for the mean, and 18.8 and 14.8 days for the median.

Signal deconvolution provides a method to infer the real number of infections using reported deaths and a time-to-death distribution. According to Gostic *et al.* [32], deconvolution is more accurate than convolving the observation time series with the reversed (backward) delay distribution. This is because backward convolution unrealistically spreads out infections across too many days in the past. We chose to use the Richardson-Lucy deconvolution algorithm [16] as we describe in what follows.

First, we need to model the stochastic process that determines the number of individuals $\delta(t)$ infected on day $t$ who will die due to the infection. Let $p$ be the probability of dying from a COVID-19 infection. Because $p$ is small we can model the number of infected people at day $t$ who will later die due to the infection as a Poisson variable with mean $\lambda(t) = p \times I(t)$.

To estimate the real incidence for $t_1 \leq t \leq t_2$ we have to obtain the vector of unknown parameters $(\lambda_{t_1}, \ldots, \lambda_{t_2})$. Let $(1, \ldots, N)$ be the days in which the number of deaths is available. Because most of deaths occur between 5 and 50 days after the infection, we can estimate the daily incidence curve between $t_1 = -49$ and $t_2 = N - 5$.

The procedure consists in applying an expectation maximization algorithm, which iterates in the space of parameters producing a sequence $\lambda^n = (\lambda_{t_1}^n, \ldots, \lambda_{t_2}^n)$. The initial guess $\lambda^0 = (\lambda_{t_1}^0, \ldots, \lambda_{t_2}^0)$ can be the death curve shifted back by the number of days as the peaks in the time-to-death distributions. If we indicate with $d$ the delay distribution then $d_i$ is the probability that an infected person will die after $i$ days from the infection. We also indicate with $D_i$ the number of reported deaths in day $i$. Now, we can define

$$q_j = \sum_{-j+1 \leq i \leq N-j} d_i, \tag{4}$$

which is the probability that an infected at time $j$ will die in the interval $1, \ldots, N$ and

$$D_i^n = \sum_{j<i} d_{i-j} \lambda_j^n, \tag{5}$$

which is the expected number of deaths to occur on day $i$, conditional on the parameters $\lambda^n$. Here $t_2 \leq j \leq t_2$. Then we can write

$$\lambda_j^{n+1} = \frac{\lambda_j^n}{q_j} \sum_{i>j} \frac{d_{i-j} D_i}{D_i^n}. \tag{6}$$

For the $n$th iteration the expected number of deaths on day $i$ is $E_i^n = \sum_{j<i} \lambda_j^n d_{i-j}$. If $\lambda^n$ were the true parameters $D_i$ would be Poisson distributed with mean $E_i^n$ and the expectation $E\left(\frac{(E_i^n - D_i)^2}{E_i^n}\right)$ would be 1. Therefore, as suggested in [16] it is more convenient to stop the iteration when the normalized $\chi^2$ statistic

$$\chi^2 = \frac{1}{N} \sum_i \frac{(E_i^n - D_i)^2}{E_i^n}, \tag{7}$$

is less than 1 for the first time.

Once the correct parameters are obtained, choosing $p$ as the IFRs reported in [28] we infer the incidence curve $I(t)$.

We note that, within this approach, the delay between infection and detection has relatively mild effects. Although cases are detected a few days after infection, if the time between

infection and detection is sufficiently narrowly distributed, then both curves are just shifted in time. The need for a time correction would be evident if the curves $C(t)$ and $I(t)$ used for training were shifted. We do not implement any such shifting, because our $C(t)$ and our $I(t)$ estimated by deconvolution are not significantly shifted.

## Bayesian machine scientist

The Bayesian machine scientist [22, 23] is a symbolic regression algorithm that, in the present context, samples closed-form models for the reporting exponent $n_i(T, C)$ from the posterior distribution $P(n_i|D)$, which gives the probability of model $n_i(T, C)$ being the one that truly generated the data $D$, conditioned on the data. The posterior is

$$
\begin{aligned}
P(n_i|D) &= \int_\Theta P(n_i|D, \theta_i)d\theta_i \\
&= \frac{1}{P(D)} \int_\Theta P(D|n_i, \theta_i)P(\theta_i|n_i)P(n_i)d\theta_i \\
&= \frac{e^{-\mathcal{L}(n_i)}}{Z(D)},
\end{aligned}
\tag{8}
$$

where $Z(D) \equiv P(D)$ is the evidence, $\theta_i$ are the parameters of the model, and the integral is over the space $\Theta$ of possible values of the parameters. The description length $\mathcal{L}(n_i)$ in Eqs (3) and (8) can be decomposed in two terms

$$
\mathcal{L}(n_i) = \mathcal{L}_L(D|n_i) + \mathcal{L}_P(n_i) .
\tag{9}
$$

Here, $\mathcal{L}_P(n_i) \equiv -\log P(n_i)$ is the contribution of the prior over models to the description length, which encapsulates our expectations about models prior to any observation of data. Following Ref. [22], we set this prior distribution using a maximum entropy principle.

The other term in the description length is

$$
\mathcal{L}_L(n_i) \equiv -\log \left[ \frac{1}{Z} \int_\Theta P(D|n_i, \theta_i)P(\theta_i|n_i)d\theta_i \right],
\tag{10}
$$

that is, the integrated likelihood encapsulating the contribution of the goodness of fit of model $n_i(T, C)$ to the description length. Since, in general, this integral cannot be calculated exactly, we use the Laplace approximation to obtain an analytical result. This approximation assumes that the likelihood $p(D|n_i, \theta_i)$ is peaked around the maximum likelihood estimators of the parameters $\theta_i^*$, and leads to [22]

$$
\mathcal{L}_L \approx -\frac{B(D, n_i)}{2},
\tag{11}
$$

where $B(D, n_i)$ is the Bayesian information criterion (BIC), defined as

$$
B(D, n_i) = -2 \log P(D|n_i, \theta_i)|_{\theta_i^*} + L \log N .
\tag{12}
$$

Here, $L$ is the total number of parameters in $n_i$ and $N$ is the size of the dataset $D$.

When looking for a single model $n(T, C)$ that fits the data of all countries simultaneously, the above arguments remain identical, except for the fact that $\mathcal{L}_L(n)$ becomes the sum of the individual description lengths of each country [23].

From Eqs (3) and (8) it is clear that the model that maximizes the posterior $p(n_i|D)$ (the most plausible model given the data) is also the model that minimizes the description length [25, 33]; that is, that the best model is the model that better compresses the data and the model.

## Model ensemble predictions and median predictive model

We use a Markov chain Monte Carlo to sample the posterior distributions $P(n_i|D)$ for the national ensemble of models (NEM; models for each country separately), and $P(n|D)$ for the global ensemble of models (GEM; global models for all countries). We sample the models using data from the training period August 2020 to November 2020 obtaining a set of models for each country $\{n_i^\alpha(T, C)\}$ for the NEM and a set of models $\{n^\alpha(T, C)\}$ for the GEM. From each sample, in Figs 2 and 4 we plot the model ensemble prediction and the median predictive model, as described next.

We compute model ensemble predictions (full lines in Figs 2 and 4) as follows. For each sampled model $\alpha$, we compute the predicted exponent at time $t$. Then, we compute the median prediction over sampled models

$$n_i^{\mathrm{NEM}}(T(t), C(t)) = \mathrm{Median}_\alpha(n_i^\alpha(T(t), C(t))) \tag{13}$$

for the NEM, and

$$n_i^{\mathrm{GEM}}(T(t), C(t)) = \mathrm{Median}_\alpha(n^\alpha(T(t), P(t); \theta_i)) \tag{14}$$

for the GEM.

We obtain the predictions of the median predictive model (dashed lines in Figs 2 and 4) as follows. In the national ensemble, for each country we select the individual model, among those sampled, that most resembles the median prediction of the ensemble in terms of the mean absolute error; this is the country's median predictive model of the NEM (MPNEM). Similarly, we select the individual model that most resembles the median predictions of the global ensemble in terms of the mean absolute error; this is the median predictive model of the GEM (MPGEM).

## Validating the Bayesian machine scientist through compartmental epidemic models

To test the ability of the BMS to recover the true incidence from data on reported cases and tests, we simulated epidemic scenarios using the well-known susceptible-infectious-recovered (SIR) model (Kermack and McKendrick [34]). The SIR model is a fundamental framework in epidemiology used to describe the spread of infectious diseases within a population; it partitions the population into three compartments: susceptible $S$, who can contract the disease; infectious $I$, who are currently infected and can transmit the disease; and recovered $R$, who have gained immunity or died, thus no longer participating in disease transmission.

The dynamics of the SIR model are governed by a set of differential equations:

$$\frac{dS}{dt} = -\beta SI,$$

$$\frac{dI}{dt} = \beta SI - \gamma I,$$

$$\frac{dR}{dt} = \gamma I.$$

Here, $\beta$ represents the transmission rate, which quantifies the rate at which susceptible individuals contract the disease upon contact with infectious individuals. The parameter $\gamma$ is the recovery rate, which describes the rate at which infectious individuals recover and move to the recovered compartment. The product $\beta SI$ represents the rate of new infections, assuming

**Table 2. Parameter values for different epidemiological scenarios.** The compartments $S$, $I$ and $R$ express fractions of the total population (100000).

|  | $\beta_0$ | $\beta_1$ | $\gamma$ | $S(0)$ | $I(0)$ | $R(0)$ |
|---|---|---|---|---|---|---|
| Single wave | 0.2 | 0 | 0.1 | 0.799 | 0.001 | 0.2 |
| Growth Phase | 0.15 | 0 | 0.1 | 0.799 | 0.001 | 0.2 |
| Decline Phase | 0.15 | 0 | 0.1 | 0.785 | 0.015 | 0.2 |
| Two waves | 0.75 | 0.3 | 0.3 | 0.796 | 0.004 | 0.2 |

**Table 3. Models and parameters to generate Tests and Reported Cases.** We indicated with $\tau$ the total number of days considered in the simulations.

| Tests | Reported Cases |
|---|---|
| $T = f_T I, f_T = 0.5$ | $C = f_C I, f_C = 0.2$ |
|  | $C = aT^n I, \quad a = \frac{0.2\,\tau}{\sum_t T^n(t)}, \quad n = 0.5, 1, 2$ |
|  | $C = T^{-b \log C} I, b = 0.03$ |
| $T = A(1 - e^{-I/B}), A = 0.15, B = 0.3$ | $C = T^{-b \log C} I, b = 0.04$ |

homogeneous mixing in the population. The term $\gamma I$ represents the rate at which infectious individuals recover.

We chose different combinations of initial conditions and parameters to simulate epidemiological scenarios similar to those observed in the data. Specifically, we simulated: (i) a growing phase, (ii) a decreasing phase, (iii) a complete epidemic wave, and (iv) two successive waves. To reproduce the two waves in the last scenario, we introduced a temporal dependency in the transmission parameter, following Mummert et al. [35]

$$\beta(t) = \beta_0 + \beta_1 \cos\left(2\pi t/365\right), \tag{15}$$

where setting $\beta_1$ to zero, we recover the old constant transmission rate $\beta_0$. In Table 2 we detailed the chosen values of the parameters and initial conditions.

Succesively, the SIR model can be modified by adding or removing compartments within the population. Several studies have simulated testing processes and the subsequent registration of cases [7, 8, 36, 37]. In our work, we adopted a similar approach, assuming that the daily number of tests administered to the population and the subsequent number of detected cases are two different fractions of the true incidence on the same day, expressed as $T = f_T I$ and $C = f_C I$. Initially, we considered $f_T$ and $f_C$ as constants. For $f_C$, we also explored the possibility of it being a function of $T$ and $C$. Specifically, to align with our case study, we modeled $f_C$ as a power law of the daily tests, $f_C = aT^n$, for two values of the exponent $n = 1$, and $2$, with $a$ constant factor. Additionally, we examined scenarios where the exponent is not constant, but has a logarithmic dependency on the number of reported cases, $T^{-b \log C}$, with $b$ constant. Furthermore, for this last case, we considered the daily number of tests not as a constant fraction of the incidence but following $T = A(1 - e^{-I/B})$, that is, an initial linear dependency, eventually saturating as the incidence increases. In Table 3 we summarize the investigated scenarios with the corresponding parameters.

## Author Contributions

**Conceptualization:** Alex Arenas, Marta Sales-Pardo, Roger Guimerà.

**Data curation:** Oscar Fajardo-Fontiveros, Mattia Mattei, Giulio Burgio.

**Formal analysis:** Oscar Fajardo-Fontiveros, Mattia Mattei, Giulio Burgio, Clara Granell, Sergio Gómez, Alex Arenas, Marta Sales-Pardo, Roger Guimerà.

**Funding acquisition:** Alex Arenas, Marta Sales-Pardo, Roger Guimerà.

**Investigation:** Oscar Fajardo-Fontiveros, Mattia Mattei, Giulio Burgio, Clara Granell, Sergio Gómez, Alex Arenas, Marta Sales-Pardo, Roger Guimerà.

**Methodology:** Oscar Fajardo-Fontiveros, Mattia Mattei, Giulio Burgio, Alex Arenas, Marta Sales-Pardo, Roger Guimerà.

**Software:** Oscar Fajardo-Fontiveros, Mattia Mattei, Giulio Burgio, Roger Guimerà.

**Supervision:** Alex Arenas, Marta Sales-Pardo, Roger Guimerà.

**Validation:** Oscar Fajardo-Fontiveros, Marta Sales-Pardo.

**Visualization:** Oscar Fajardo-Fontiveros, Marta Sales-Pardo, Roger Guimerà.

**Writing – original draft:** Oscar Fajardo-Fontiveros, Mattia Mattei, Giulio Burgio, Clara Granell, Sergio Gómez, Alex Arenas, Marta Sales-Pardo, Roger Guimerà.

**Writing – review & editing:** Oscar Fajardo-Fontiveros, Mattia Mattei, Giulio Burgio, Clara Granell, Sergio Gómez, Alex Arenas, Marta Sales-Pardo, Roger Guimerà.

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
