## [Decision Letter · Decision Letter 0]

29 Mar 2024

Dear Dr. Guimerà,

Thank you very much for submitting your manuscript "Machine learning mathematical models for incidence estimation during pandemics" for consideration at PLOS Computational Biology.

As with all papers reviewed by the journal, your manuscript was reviewed by members of the editorial board and by several independent reviewers. In light of the reviews (below this email), we would like to invite the resubmission of a significantly-revised version that takes into account the reviewers' comments.

I agree with both reviewers that this is a potentially interesting method, but substantially more work is needed to demonstrate a number of key claims made in the manuscript. These include: the ability to estimate real-time incidence for control, extensibility to other respiratory pathogens, limitations (especially as it pertains to under-reporting), and justification of the black-box approach. In many cases, it would be sufficient to simply temper claims made in the paper; however, I believe it's critical that the authors perform additional computational experiments to provide meaningful demonstration of the methods abilities (and to justify aspects of the approach). I also agree with the reviewers that more work is needed placing this approach in the context of existing models and expanding/justifying the Bayesian set up discussed. I hope that the reviewers pay careful attention to the thoughtful, detailed comments from the reviewers during their revision.

Please also ensure that your code is made available in compliance with our code-sharing policy.

We cannot make any decision about publication until we have seen the revised manuscript and your response to the reviewers' comments. Your revised manuscript is also likely to be sent to reviewers for further evaluation.

Sincerely,

Samuel V. Scarpino

Academic Editor

PLOS Computational Biology

Virginia Pitzer

Section Editor

PLOS Computational Biology

I agree with both reviewers that this is a potentially interesting method, but substantially more work is needed to demonstrate a number of key claims made in the manuscript. These include: the ability to estimate real-time incidence for control, extensibility to other respiratory pathogens, limitations (especially as it pertains to under-reporting), and justification of the black-box approach. In many cases, it would be sufficient to simply temper claims made in the paper; however, I believe it's critical that the authors perform additional computational experiments to provide meaningful demonstration of the methods abilities (and to justify aspects of the approach). I also agree with the reviewers that more work is needed placing this approach in the context of existing models and expanding/justifying the Bayesian set up discussed. I hope that the reviewers pay careful attention to the thoughtful, detailed comments from the reviewers during their revision.

Reviewer's Responses to Questions

**Comments to the Authors:**

Reviewer #1: Review: PCOMPBIOL-D-23-02094

This paper attacks a key problem in tracking infectious diseases – how can we correct for the biases in reported cases (which are often incomplete representations of infections) to better estimate the actual infections circulating in the population. While I do think this study has some potential, there is not sufficient evidence currently to support its main claims. Specifically, I am not convinced that the proposed machine learning method has been shown to improve what is already known from the literature and I think that results have not been demonstrated to be as general as is sometimes claimed. Some substantial additional work is required.

Major comments

1. As presented the method is somewhat opaque and not sufficiently explored for what I believe is its first application in epidemiology. It is unclear how it performs when incidence is low or high or how it might depend on having a high-quality estimate of the IFR (and on if the IFR is large or small). An extensive simulation study looking at these points with a known ground truth and also better quantifying and calibrating predictive performance against something simple like the ratio of C(t)/T(t) and an existing statistical method (see point 4) is needed. The ground truth could examine several closed-form T(t) models including the constant and more complex ones. These simulations will help clarify and provide insight into what value the proposed methodology adds.

2. Using deconvolution with the IFR to obtain a benchmark may be misleading on real data (which is also why point 1 is important). The IFR may change with time and phase of the epidemic. Assuming a fixed IFR might then suggest those changes have to be accounted for within the exponent inferred when training the method. This might affect conclusions about whether n(T, C) is ‘constant’. This confounder is worth testing and explaining. More generally, the strengths, limits and assumptions of the approach are not well explored and it is hard to gauge its benefits and critique its relative value. More analyses towards this direction would make the manuscript more convincing.

3. If we require the IFR or death time series to train this algorithm then what is the benefit over simply performing deconvolution and estimating infections that way? This was done with success by several groups starting from the Goldstein et al paper already cited and also https://www.nature.com/articles/s41586-020-2405-7 and a number of other papers. This common approach (there are 10s of papers similarly converting deaths to infections) needs to be discussed and the proposed methodology has to add value relative to them. There are also a number of ‘nowcasting’ approaches that are also worth considering.

4. There is little engagement with the literature on the subject of under-reporting and the methods used to understand, correct and quantify its impact. This paper needs to be contextualised against those works to clarify its contribution. Some studies on methods and how performance depends on reporting variability that should be discussed:

a. https://journals.plos.org/plosntds/article?id=10.1371/journal.pntd.0006161

b. https://academic.oup.com/aje/article/190/9/1908/6217341?login=false

c. https://www.nature.com/articles/s41586-020-03095-6

d. https://www.nature.com/articles/s43588-022-00313-1

e. https://onlinelibrary.wiley.com/doi/10.1002/sim.6015

Not only do these contain the common methodology for correcting under-reporting they also discuss how the form of the under-reporting (i.e., how time-varying it is) affects our estimation and prediction performance.

5. Not enough motivation is provided for why machine learning is needed and what is adds when solving this problem. The stated benefits seem to be accounting for more complex n(T, C) formula and getting closed-form descriptions. Yet the latter, which is meant to help interpretability, are not written explicitly nor are consequences explored. Additionally, the variable of actual interest is 1/b and methods exploring complex 1/b forms do exist so more work is needed in this direction to distinguish what machine learning is bringing.

6. There is no mention or attempt to account for delays in reporting cases (which are not negligible). These also add temporal components to the reporting rate (see again some of a-e above) and the impact of this related noise source needs to be explored for any analysis on real data. Delays in reporting may be adding some of the complexity to n(T, C) and are extremely important for correcting cases to infections (also see nowcasting). A related paper is https://onlinelibrary.wiley.com/doi/abs/10.2307/3315826.n1

7. The value of using the minimum description length (MDL) is unclear. With the Laplace approximation it becomes a BIC with a prior penalty. How would performance fair if we simply use the BIC? Other papers apply MDL to epidemics via Fisher approximations (https://academic.oup.com/sysbio/article/69/6/1163/5825296?searchresult=1) or proper scoring rules (https://www.ncbi.nlm.nih.gov/pmc/articles/PMC6386417/), which amount to the equivalent. These also need to be discussed for context else the results about parsimony do not seem that new and are hard for unaware readers to parse.

8. It would be good to convert the exponents in Fig 1 back to the actual rate of reporting (1/b, which is what is really of interest) and in Fig 2 to include C(t) for reference.

9. Last, is there any data for the case studies chosen to show what the best estimate of the infections in those regions are (versus the deconvolution that depends on IFR)? There should be for France at least. If it can be shown relative to those best estimates that this approach outperforms others as noted also in earlier points, this would strengthen the paper and help with interpretation of the Figs as e.g., it is unclear if the estimates for UK, Hungary, France and Catalonia are that good in absolute terms.

Reviewer #2: This is a review of "Machine learning mathematial models for incidence estimation during pandemics" from Fajardo-Fontiveros and colleagues, submitted to PLOS Computational Biology. The big idea in this paper is to use Bayesian symbolic regression (which the authors call a Bayesian Machine Scientist, BMS) to learn not only parameters (as we might in a typical Bayesian modeling problem) but a function as well. While BMS is described elsewhere, here the authors ask: can we use it to learn about the underreporting of COVID-19 infections? They express underreporting as a factor b, defined as the fraction of total infections (I) which are reported as cases (C), and the BMS specifically provides a posterior over functions describing how b scales with the fraction of the population being tested.

The technical details of how the BMS works are provided (and developed in detail in Ref 15), but for the purposes of this paper, the authors really do two things: First, they fit both nation-specific and global models and ask which provides a better fit after accounting for model size (via rigorous model selection). [Answer: a global model is more parsimonious.] Second, they show that (1) the relationship between the underreporting factor b and the amount of testing T varies over time, and (2) a time-varying relationship between b and T leads to good predictions of incidence. There are a few issues with the paper that hold it back, which I think could be addressed, but might require more changes than the authors had in mind. They follow, below.

1. Fundamentally, this paper is framed about solving one problem, but then goes on to solve a different problem. What I mean is: the authors make a reasonable case that this method provides a means to retroactively adjust case counts to estimate true incidence. However, the framing of the paper is about real-time measurement of incidence for the purposes of control. I don't see results that evaluate a real-time prediction. Instsead, they use 4 months of training data (Aug, Sept, Oct, Nov 2020) to predict two months of test data (Dec 20, Jan 21). The models also require mortality, case, and testing (T) data on which to train, but I don't think these are actually available in real time.

2. Would this work on other infectious diseases, or ILI? The Goldstein deconvolution approach was developed for flu if I recall correctly... In any case, if the authors hope to argue that the BMS approach here is going to be useful for future pandemic control, then it would be great to be able to make that case on less time-restricted and pathogen/variant-restricted data.

3. What are the limitations of this approach, data, conclusions, etc? Relatedly, is Goldstein really the right ground truth? Are there alternatives? How could this model work on *actual* real time data, knowing that many of the datastreams during the acute phase of the pandemic were delayed by varying amounts (or published by reporting data -vs- swab date, etc)?

Relatedly, are there other extensions that could be discussed?

4. Are the functional forms in the country specific OR global models interesting or useful? Is the composition of each country's posterior the same, in terms of the functional forms we see? In other words, can we learn anything about the actual samples from the BMS posterior that teach us something about surveillance, infectious diseases, testing, etc? Or is this [simply] a powerful black-box method, and we shouldn't try to look under the hood to learn more? As written on Line 122, the model selection findings, showing that a common mechanism for underreporting is the more parsimonous model, raise "questions about the nature of such mechanisms," but the authors have missed an opportunity to help answer or discuss said mechanisms.

In conclusion, I enjoyed reading this paper, and it's a nice idea with some new technology for the readership of PLOS Comp Bio. But, the paper feels underdeveloped. We learn that a previously developed hammer (BMS) can hit this particular nail (2020-2021 COVID-19 undereporting), but don't learn much about the hammer, this class of nail, other similar nails, and so on. The paper has a lot of potential, but without unpacking more about what the BMS *means*, and what it *tells us*, it could easiily be any other black-box ML approach--with the exception of the model selection result. Unfortunately, the model selection result (one model, not lots of country-specific models) is not enough, in my opinion, to hang the paper on.

**Have the authors made all data and (if applicable) computational code underlying the findings in their manuscript fully available?**

Reviewer #1: **No: **I could be wrong but did not see code for reproducing the analyses. They do mention where the raw data is taken from.

Reviewer #2: **No: **Is the code for the BMS available? Or code for reproducibility? The availability statement mentions the data, but I don't see a repository associated with this paper. Please forgive me if I've overlooked it!

PLOS authors have the option to publish the peer review history of their article (what does this mean?). If published, this will include your full peer review and any attached files.

Reviewer #1: No

Reviewer #2: No
---

## [Decision Letter · Decision Letter 1]

27 Sep 2024

Dear Dr. Guimerà,

Thank you very much for submitting your manuscript "Machine learning mathematical models for incidence estimation during pandemics" for consideration at PLOS Computational Biology. As with all papers reviewed by the journal, your manuscript was reviewed by members of the editorial board and by several independent reviewers. The reviewers appreciated the attention to an important topic. Based on the reviews, we are likely to accept this manuscript for publication, providing that you modify the manuscript according to the review recommendations.

Sincerely,

Samuel V. Scarpino

Academic Editor

PLOS Computational Biology

Virginia Pitzer

Section Editor

PLOS Computational Biology

Reviewer's Responses to Questions

**Comments to the Authors:**

Reviewer #1: Review: PCOMPBIOL-D-23-02094R1

Many thanks to the authors for a detailed and meticulous reply. I think the manuscript is much improved. However, I am still left unconvinced about the added value of the approach. Some reasonable additions to the new simulations and some qualifications could resolve this.

Fig R1 – I like this figure but:

1) Please include C(t)/I(t) where I(t) is either from the simulation or estimated from deaths. In short while the shown predictions do look good, without a reference for comparison it is hard to understand if they are improved in some sense.

2) Following on from that, a comparison against another method e.g. maybe just a simple statistical time series model. It is not that I disbelieve the approach, just that I want to understand what it improves upon to justify the increased complexity.

The authors call these real-time estimates. But if you require 150 days for training, it is only real time after quite a substantial ‘burn-in’. After 150 days it may be too late for these analyses. Can the authors qualify what minimum level of training is needed or explain this issue?

Is the Chiu et al method simply the constant exponent? It isn’t made that clear what they do and if the authors base their results on being more accurate than this, then it should be made very clear what Chiu et al propose. In the paper Chiu et al is only mentioned once. Are Chiu et al using symbolic regression or another method? Please clarify.

As the authors note any reasonable estimate of incidence is needed for training and hence this does not depend on good IFR estimates and deconvolution. I have no issue with that, my concern was that if I had a good IFR and inferred infections from it (which is now a common approach) then what benefit do I gain from applying the authors’ method? This is the question that still needs addressing.

The authors state that delays in reporting are not a concern. The field of nowcasting exists precisely because it is a notable problem (and even more substantial one for deaths), even in high-income countries. While reported cases appear daily, as the authors note, the cases that are reported are not all the cases that should be counted on that day. This is due to such delays, which for COVID were above 5 days. Please ensure the text acknowledges this issue.

Fig R2 – I am unsure what the estimates being compatible means.

Last, what is the interpretation behind why the authors believe a single model across countries is beneficial over the multiple models? This could be useful for readers.

Reviewer #3: Authors are advised to include the recommendation of future work.

Reviewer #4: The points raised by the reviewers in the first round of the review have been satisfactorily addressed in the current version of the manuscript. The manuscript now provides stronger evidence for the robustness, accuracy, and it engages more deeply with relevant literature, improving its clarity and theoretical grounding.

The introduction offers a comprehensive background, particularly focusing on the challenges of under-reporting during pandemics, and how this impacts real-time incidence estimation. It outlines the motivation for developing a machine learning model to correct under-reporting based on reported cases and testing data.

The reasons for performing the study are clearly defined and the study objectives are clearly articulated. The methods used in the study are generally appropriate for the objectives of estimating real-time incidence during pandemics. The use of Bayesian symbolic regression and machine learning techniques to derive closed-form mathematical models is well-suited to address the challenge of under-reporting, and the study's focus on testing the model across various pandemic scenarios (using synthetic and real-world data) is robust. Additionally, the choice to explore whether a single unified model can apply across multiple countries seems innovative and relevant to the study’s goals of generalizability.

The manuscript provides much of the necessary information for replication, including details of the datasets used, the Bayesian Machine Scientist (BMS) method, and the epidemic models tested in the simulation study. However, some of the technical details regarding model selection, training procedures, and hyperparameter choices for the machine learning algorithms could be expanded. For instance, a more detailed description of the parameters used in the Bayesian symbolic regression or the specific implementation details of the epidemic models (e.g., the equations governing the synthetic data generation) would ensure that a researcher can fully replicate the experiments. Additionally, including code or detailed pseudocode could further improve reproducibility.

The manuscript includes appropriate references when discussing established methods for under-reporting correction and machine learning. The authors refer to key works, such as Chiu et al. and the Goldstein deconvolution approach, and they also cite relevant literature on Bayesian symbolic regression. The manuscript includes mean absolute error (MAE) metrics to quantify prediction errors, which are easy to interpret.

The manuscript touches on some limitations, such as the reliance on reported case and testing data and the assumption that the infection fatality rate (IFR) remains constant over time. However, the discussion of limitations could be expanded. Major limitations that should be addressed include: (i) potential overfitting to specific pandemic scenarios; (ii) The implications of incomplete or delayed testing data.

Based on the review provided, the following points should be explored in this new round of revisions:

1. Clarification of Technical Details for Replication;

2. Expansion of Limitations;

3. Additional Reproducibility Considerations;

After these points are addressed, the reviewer recommends that the manuscript be accepted for publication.

Reviewer #5: The revised manuscript presents an approach utilizing Bayesian symbolic regression to estimate the real-time incidence of infectious diseases, focusing on COVID-19. Below is a detailed review of the key strengths and areas for improvement in this second-round submission.

Strengths:

1. The manuscript addresses a critical issue in epidemic management: underreporting due to limited testing, especially during pandemics. The potential application to other infectious diseases strengthens the paper's relevance beyond the immediate context of COVID-19.

2. A comprehensive simulation study evaluating the approach across different epidemic scenarios (e.g., growing, decreasing, and two-wave epidemics) provides solid evidence for the method's validity. The BMS's ability to recover the true reporting model across varying conditions is an important contribution.

Drawbacks:

1. While the novel application of symbolic regression is a strength, the manuscript would benefit from further clarification on what unique contributions this method brings beyond existing nowcasting and underreporting correction approaches.

2. One limitation noted in the manuscript is the difficulty interpreting the closed-form expressions derived by BMS. The paper mentions that efforts to interpret these expressions were not fruitful due to parameter variability across countries. While interpretability is not always required for machine learning models, symbolic regression is often praised for this ability, which is not emphasized here.

3. A previous review raised concerns about reporting delays, which are not sufficiently addressed in the current version. While the authors argue that these delays were not major during the COVID-19 pandemic, they should still be explored.

Recommendations:

1. Expand the discussion on BMS's relative advantages over traditional deconvolution methods. Highlight BMS's unique interpretative power or computation speed.

2. Consider providing simplified examples of interpretable symbolic regression outputs, even if these are less complex models. This would help demonstrate the potential for insight extraction beyond predictive power. If interpretability is not a central goal, this should be clearly stated.

3. Provide a more detailed discussion of how reporting delays (both in test results and case reports) could influence the results and whether corrective strategies, such as time-shifting the data.

**Have the authors made all data and (if applicable) computational code underlying the findings in their manuscript fully available?**

Reviewer #1: **No: **

Reviewer #3: Yes

Reviewer #4: Yes

Reviewer #5: None

PLOS authors have the option to publish the peer review history of their article (what does this mean?). If published, this will include your full peer review and any attached files.

Reviewer #1: No

Reviewer #3: **Yes: **Elayaraja Aruchunan

Reviewer #4: **Yes: **Gustavo Barbosa Libotte

Department of Computational Modeling, Polytechnic Institute, Rio de Janeiro State University, Nova Friburgo, Brazil

Reviewer #5: No

Figure Files:

Data Requirements:

Reproducibility:

References:

---

## [Editor Report · Decision Letter 2]

2 Dec 2024

Dear Dr. Guimerà,

We are pleased to inform you that your manuscript 'Machine learning mathematical models for incidence estimation during pandemics' has been provisionally accepted for publication in PLOS Computational Biology.

Best regards,

Samuel V. Scarpino

Academic Editor

PLOS Computational Biology

Virginia Pitzer

Section Editor

PLOS Computational Biology

Feilim Mac Gabhann

Editor-in-Chief

PLOS Computational Biology

Jason Papin

Editor-in-Chief

PLOS Computational Biology

---

## [Editor Report · Acceptance letter]

12 Dec 2024

PCOMPBIOL-D-23-02094R2 

Machine learning mathematical models for incidence estimation during pandemics

Dear Dr Guimerà,

I am pleased to inform you that your manuscript has been formally accepted for publication in PLOS Computational Biology. Your manuscript is now with our production department and you will be notified of the publication date in due course.

With kind regards,

Anita Estes
